# Detection of Bering Sea Slope Mesoscale Eddies Derived from Satellite Altimetry Data by an Attention Network

Yuyuan Zhang [1,2], Na Liu [1,2,*], Zhiyuan Zhang [3], Min Liu [3], Long Fan [4], Yunbo Li [3], Lei Yang [1,2] 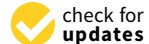, Lina Lin [1,2] and Hongxia Chen [1,2]

1 Key Laboratory of Marine Science and Numerical Modeling, First Institute of Oceanography, Ministry of Natural Resources, Qingdao 266061, China
2 Laboratory for Regional Oceanography and Numerical Modeling, Pilot National Laboratory for Marine Science and Technology, Qingdao 266237, China
3 The 91001 Unit of PLA, Beijing 100036, China
4 Navy Research Institute, Tianjin 300061, China
* Correspondence: liun@fio.org.cn

**Abstract:** In the Bering Sea slope, ocean eddies are essential physical processes that carry nutrients to the shelf. The development of the satellite altimeter has facilitated the observation of oceanic eddies. Attention networks are used as the core algorithm for eddy detection to suppress feature responses in irrelevant non-eddy areas, which can address the issue of sample imbalance in high-latitude ocean eddies. Furthermore, data from both the sea surface height (SSH) and geostrophic velocity were employed as model inputs to integrate more eddy-related properties. The results of ocean eddy detection using this method and the dataset allowed more eddies to be detected than with traditional vector geometry-based methods and only SSH-based models. This study also incorporated the results of multiple deep learning models to increase both the overall and single-day eddy detection efficiency. As a result, the algorithms in this paper show that attention networks and geostrophic velocity data are both appropriate for high-latitude ocean eddy identification. This makes a contribution to the further application of deep learning methods to satellite altimetry data.

**Keywords:** satellite altimeter; Bering sea slope; attention network; deep learning; oceanic eddy; geostrophic velocity



## 1. Introduction

Mesoscale eddies are important oceanic phenomena that occur in the upper ocean, with spatial sizes of tens to hundreds of kilometers and lifetimes spanning from tens to hundreds of days. They can be classified as cyclonic or anticyclonic eddies based on their motion patterns and as cold or warm eddies based on their thermal states [1–3]. The Bering Sea is a typical sea in which eddies frequently occur. The mechanisms that generate these eddies include geostrophic effects, instability, wind forcing, strong currents through the east pass, and topographic effects [4–8]. Anticyclonic eddies along the shelf break in the Bering Sea slope (BSs), which bring cold, nutrient-rich water to the surface from the depths, have a significant impact on the Bering Sea's main productivity, particularly the "Green Belt" [9].

A satellite altimeter gathers information about the ocean by sending a signal to the surface of the water and then analyzing the signal that comes back. Today, we have ESA-1/2, Envisat, Sentinel-3A/B, Jason-1/2/3, Sentinel-6, HY-2A/B/C/D, SARAL/AltiKa, CryoSat-2, and ICESat-1/2, which provide a homogeneous, highly accurate, long-term series of more than 30 years [10–14]. These high-quality satellite altimetry data have greatly contributed to the study of mesoscale ocean eddies [15–17].

The traditional approaches to automated eddy detection are mainly separated into physical parameter methods and geometric feature methods, such as the Okubo–Weiss

(OW), Winding-Angle (WA), and Vector Geometry (VG) methods [18–20]. However, all of these methods require contour iterations or sophisticated computing procedures and are inefficient in terms of computational efficiency. Furthermore, these methods also suffer from the necessity for artificial parameter estimation, are susceptible to SSH noise, rely on treating eddies as ellipsoidal, and necessitate contour iterations or sophisticated computer procedures [21,22].

Deep learning algorithms have been widely applied in the marine industry in recent years. Nonlinear transformations of deep learning can abstract the representations of features, providing a unique advantage in the extraction and learning of target features. A variety of deep neural networks introduced in the computer vision literature have been used to find and classify eddies by applying image processing techniques to satellite altimetry data [23]. Lguensat et al. [24] detected ocean eddies from SSH data based on the encoder–decoder network U-Net. Xu et al. [25] performed the detection of ocean eddies using the Pyramid Scene Parsing Network (PSPNet) in a semantic segmentation framework, which introduced a null-roller and a pyramid pooling model to capture more contextual relationships by making full use of global and local information and demonstrated good results in the western Pacific Ocean. Considering the effects of ocean eddies on sea surface temperatures, Liu et al. [26] proposed a multimodal U-Net eddy recognition model for eddies based on fused satellite SSH and SST data, which was significantly better than a deep learning model that only considered a single feature. In addition, network and feature inputs, as well as mesoscale eddy mask data, may have a significant impact on a model's performance. Santana et al. [27] compared multiple variations of the U-Net network and different eddy segmentation mask data in a mesoscale eddy detection task in the South Atlantic near South America and determined that the segmentation mask data had a far greater influence on detection efficiency than that of the network enhancements.

Although prior research on the identification of mesoscale eddies using deep learning algorithms has shown excellent results in a range of seas, no study has been undertaken specifically for high latitudes, where mesoscale eddies are often tiny and sample imbalances are more pronounced. In addition, eddies on the Bering Sea slope serve as an important physical process for water column and nutrient transport. It is essential to detect them accurately. Therefore, using attention networks to rectify the imbalances in eddy samples on the Bering Sea slope and achieve effective detection of mesoscale eddies on the BSs is the major objective of this study.

## 2. Data

The daily sea-level anomaly (SLA) data with a spatial resolution of $0.25° \times 0.25°$ were used to characterize the mesoscale eddies of the BSs. Daily temporal data could meet the requirement for huge samples for model training and were also enough to describe eddies that lasted for dozens of days. Similarly, a spatial resolution of 25 kilometers provided the neural network with sufficient detail about the features of eddies ranging from tens to hundreds of kilometers. The SLA data were obtained from the global multi-satellite along-orbit altimetry data product of the Copernicus Marine Environmental Monitoring Service (CMEMS). The data products combined all satellite altimetry data from 1993 to the present, including the data from TP, Janson-1/2, ERS-1/2, ENVISAT, GFO, Cryosat-2, Sara/Atika, and Haiyang-2A. The geostrophic velocity data contained zonal and meridional velocities calculated from sea surface height anomaly data. To increase the number of eddy boundary pixels and improve detection accuracy, this paper used nearest-neighbor interpolation to interpolate the SLA data and geostrophic velocity data onto a $0.125° \times 0.125°$ grid [28].

We used 10,227 days of SLA and geostrophic velocity data from 1993 to 2020 as the dataset in this study, with 3653 samples from 2011 to 2020 as the test set and 6574 samples from 1993 to 2011 as the training data (of which 30 percent was the validation set). In addition, this research used a vector geometry (VG)-based algorithm to label the eddy boundaries from the geostrophic velocity data as the labels for training. Anticyclonic eddies were labeled as 1, cyclonic eddies as 2, and land and non-eddy seas as 0.

## 3. Methods

### 3.1. Deep Learning Models Based on Attention Networks

For the effective detection of mesoscale eddies in the BSs, this study uses an attention-network-based deep learning model to simultaneously extract features from SLA data and geostrophic velocities for learning. An attention network is a network that learns a set of weighting coefficients autonomously from the network and "dynamically weights" them to emphasize areas of interest while suppressing irrelevant background areas. This study applies two typical attention networks: the channel attention network and the spatial attention network. The characteristics that are significant on whichever channel are the topic of channel focus. By learning the significance of each feature channel, channel attention may determine the interdependencies across feature channels and assign each channel a unique weight coefficient. This causes significant feature responses to stand out and causes fewer significant feature responses to disappear. Spatial attention to where features are meaningful is achieved by assigning weights to features at all locations to selectively aggregate features that are related to each other, thereby suppressing features at non-important locations [29].

Figure 1 shows the eddy detection model built in this study based on the Attention Resnet U-network (AttresU-net). The AttresU-net deep learning algorithm consists of three main modules: the attention gating module, the U-net semantic segmentation base module, and the residual module. The model introduces the residual module into the U-net to replace the convolutional structure in the original encoder, allowing the entire network to deepen to extract more mesoscale vortex features while avoiding gradient vanishing. Each residual module consists of a $3 \times 3$ convolutional layer, an IN layer, a ReLu activation function, and a short connection consisting of a $1 \times 1$ convolutional layer and an IN layer. For the encoder path, each residual module is connected to each other one by a $2 \times 2$ maximum pooling layer. For the decoder path, each stage is convolved by a $2 \times 2$ upsampling to achieve recovery of the original resolution. The attention gate (AG) module, which suppresses the responses of features in irrelevant background regions (e.g., in non-eddy seas and land), increases the sensitivity of the model to eddies with less computational effort and is central to the model. In the U-net architecture, AGs are able to filter neuronal activation during forward and backward transfer by weighting gradients from background regions, such as land and non-mesoscale eddy seas, downwards during backward transfer. As a result, the model parameters are updated in the shallower layers primarily based on spatial regions in the task [30,31].

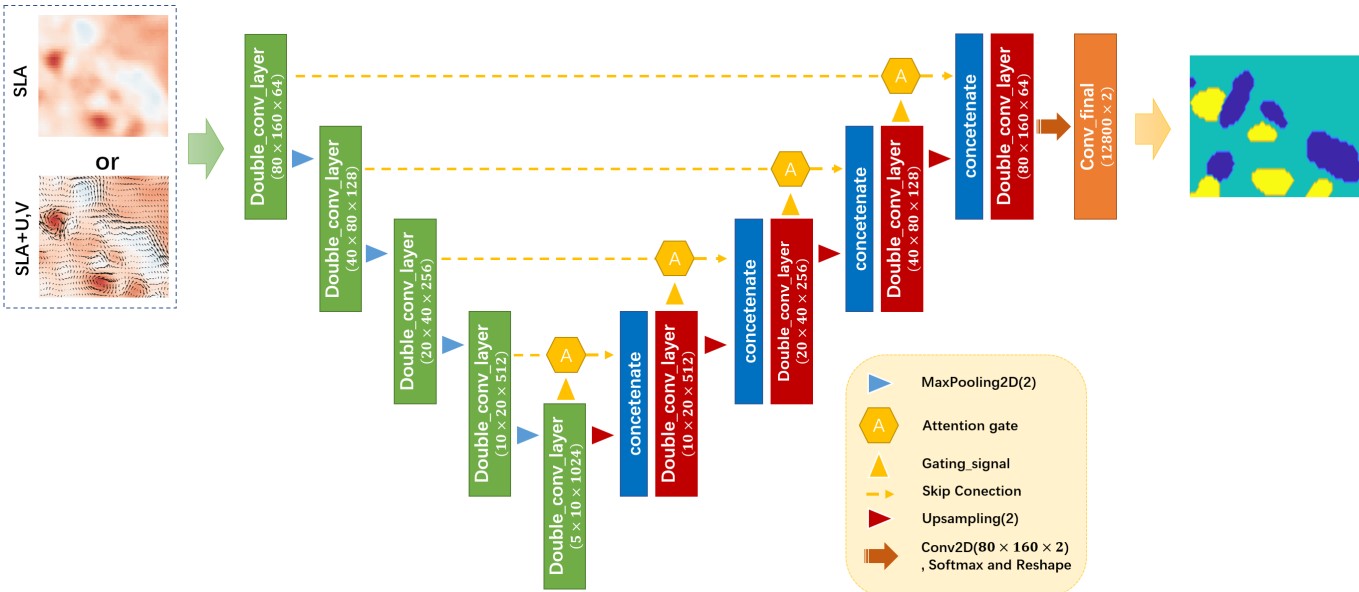

**Figure 1.** Schematic diagram of the Attention Resnet U-Network (AttresU-net).

Figure 2 shows an eddy detection model based on the Dual Attention Network (Danet). Danet's backbone network was Dilated RestNet, with the downsampling operations removed and dilated convolution used in the last two RestNets to obtain a feature mapping of 1/8 of the size of the inputs. The attention module was then supplied with the feature map output by the backbone network. Then, the reshaped D was multiplied with the spatial attention map, multiplied by the scale factor $\alpha$, reshaped again to its original shape, and, finally, added to A to obtain the final output feature map E ($10 \times 20 \times 512$). The processing of the channel attention module was similar, except that the opposite matrix multiplication was used to obtain a channel attention map of ($512 \times 512$). The outputs of the two channel attention modules were elementally summed to complete the feature fusion and, finally, convolved to produce a prediction map [32].

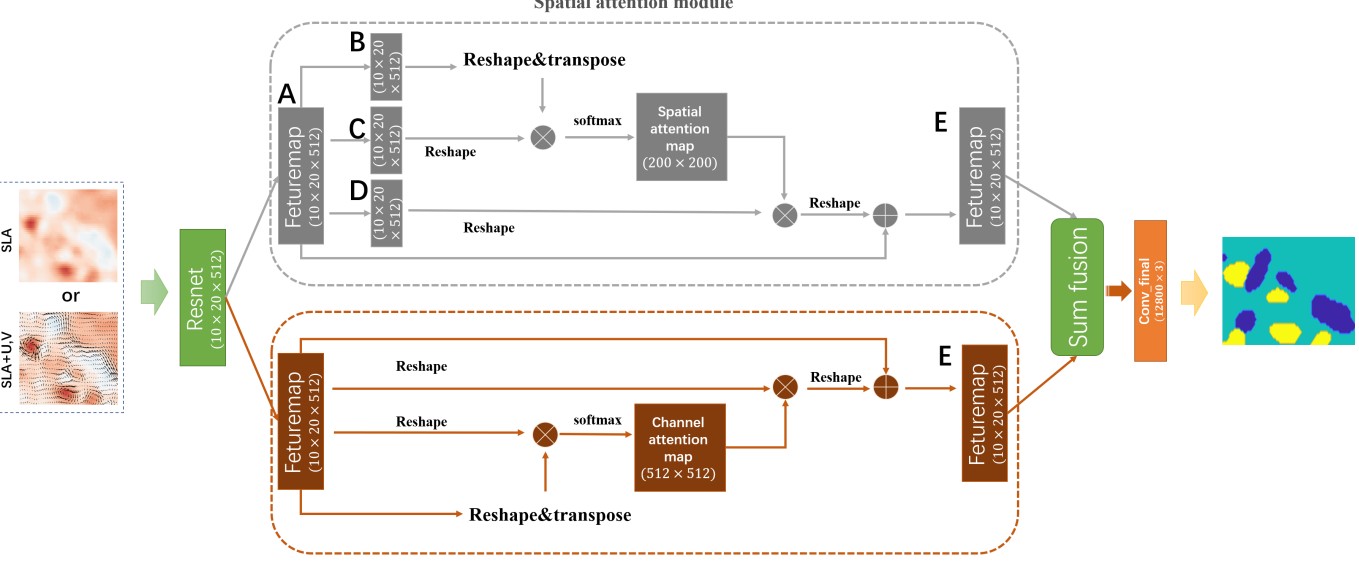

**Figure 2.** Schematic diagram of the Dual Attention Network (Danet).

### 3.2. Experimental Setup

The mesoscale eddy detection model was trained using the Adam optimizer with the maximum learning rate set to 0.001 and the learning rate adjustment scheme set to use the ReduceLROnPlateau method. After 5 epochs, the learning rate was halved without any further improvement in the validation dataset loss function. The terminating conditions looked for 10 consecutive epochs without any improvement in the loss function; if this condition was fulfilled, we considered the optimal value to be attained and halted the process. The Dice coefficient, which is commonly used in semantic segmentation tasks, was chosen as the metric, since it was unaffected by the fraction of foreground. In addition, accuracy was used as a metric to evaluate whether the eddy classification was accurate. These metrics are defined as follows:

$$Dice = \frac{2|P \cap T|}{|P| + |T|}, \tag{1}$$

where $p$ (prediction) is the prediction graph for a particular classification, and $T$ (true) is the true distribution of this classification. $|p|$ or $|T|$ represents the number of elements in the matrix.

$$acc = \frac{TP + TN}{TP + TN + FP + FN}, \tag{2}$$

where *TP* (true positive) is the number of eddies that are correctly detected, *FP* (false positive) is the number of eddies that are over-detected, and *TN* (true negative) and *FN* (true negative) indicate the numbers of points properly and incorrectly recognized as background. The models were built with the Keras framework and trained on an Nvidia GeForce RTX3090 GPU.

## 4. Results

### 4.1. Eddy Detection

In this study, SLA and geostrophic velocity data from the altimetry data were used to train three different models to detect mesoscale eddies in the Bering Sea slope region. AttresUnet, Danet, and Eddynet comprised the three models. The Eddynet model was proposed by Lguensat and was used as a reference model in this study [24].

Table 1 compares the performances of different training datasets and different models for eddy detection in the BSs area according to the evaluation metrics. The evaluation metrics Dice coef anti and Dice coef cyc represent the similarity of the model's recognition profile to the ground truth for anticyclonic eddies and cyclonic eddies, respectively, the Mean Dice coef is the average similarity including the non-eddy region, and Accuracy represents the accuracy of the classification. In order to evaluate the model training effect more realistically, the model was tested in this paper by using data from the decade 2011–2020. When the U and V components of the geostrophic velocity were entered into the model together with SLA as different channels, all the metrics tested by the model were very significantly improved compared to those obtained when using only SLA data. This may have been due to the fact that the geotrophic velocity field itself was representative of the spatial characteristics of the eddies. When we examined the evaluation results of the different models, we noticed that the evaluation indexes of AttresUnet were slightly better than those of Eddynet. The AttresUnet model introduced an attention network module to its structure, which helped to ease the sample imbalance problem to some extent. The Danet model had the best metrics among the three models, with an average similarity of 0.74–0.77 with the ground truth and a classification accuracy of 91.88–93.75%. Unlike AttresUnet, which inserted the attention mechanism module into U-net, Danet directly used the attention network to extract eddy features for output classification. These results indicate that the attention network was more competent in training the eddy detection model.

**Table 1.** Metrics derived from the models trained on two datasets.

|  | Dice Coef Anti | | Dice Coef Cyc | | Mean Dice Coef | | Accuracy (%) | |
|---|---|---|---|---|---|---|---|---|
|  | SLA | SLA,U,V | SLA | SLA,U,V | SLA | SLA,U,V | SLA | SLA,U,V |
| AttresUnet | 0.65 | 0.72 | 0.54 | 0.62 | 0.71 | 0.77 | 90.72 | 92.69 |
| Danet | 0.69 | 0.72 | 0.57 | 0.62 | 0.74 | 0.77 | 91.88 | 93.75 |
| Eddynet | 0.65 | 0.71 | 0.52 | 0.63 | 0.71 | 0.76 | 90.78 | 92.37 |

It is necessary to consider that in the study of eddy detection with deep learning, model evaluation results are based on the ground truth, which is often replaced by automatic recognition algorithms based on parameters and geometric contours, such as OW, VG, and Openeddy. The algorithms themselves are subject to certain errors and cannot replace the real eddy field, so the final eddy detection effect needs further analysis. Figure 3 shows the ocean eddies in the BSs identified 9 September 2016 using the VG, AttresUnet, Danet, and Eddynet algorithms with different training datasets. In the snapshot of eddies identified on 9 September 2016, all three deep learning algorithms identified more eddies than the VG algorithm in both datasets. In the ten-year dataset, it was also true that the deep learning algorithm found more eddies than the VG method did (67,205–71,548 detected by the deep learning algorithm and 64,244 by the VG algorithm). Furthermore, around 65.5°N, 177°W, there was a considerably high SLA and a closed anticyclonic velocity field. All

deep learning algorithms detected the eddy at this site, but the VG algorithm did not. In addition to missing crucial eddies, the VG algorithm also suffered from uncertainty in the depiction of eddy borders, e.g., anticyclonic eddies between 55–57°N, 175–179°W.

The differences among deep learning models are difficult to show in a few snapshots. Overall, the Danet trained with the SLA and geostrophic velocity detected the most eddies (with a number of 71,548). However, among the eddies detected by Danet, a few were precisely located but entirely mischaracterized in shape. The U-net-based AttresUnet and Eddynet algorithms, on the other hand, tended to depict shapes more accurately. Therefore, this study suggests combining eddy detection results from multiple models to produce more accurate results.

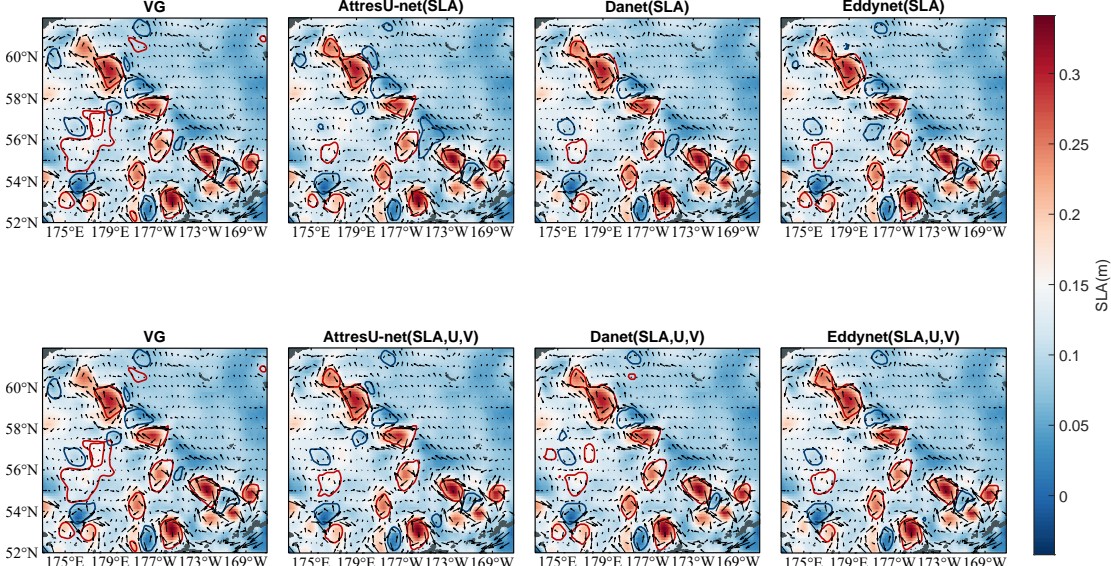

**Figure 3.** Comparison of the oceanic eddies detected by the four different algorithms in the BSs region on 9 September 2016. The upper panel shows the comparison of the VG, Attresunet, Danet, and Eddynet eddy detection results from the SLA dataset. The bottom panel shows the comparison of the VG, Attresunet, Danet, and Eddynet eddy detection results from the SLA and geostrophic velocity datasets. The shading is SLA, red contours are anticyclonic eddies, and blue contours are cyclonic eddies.

We used the results of the AttresU-net (SUV) model, which provided a more accurate depiction of the shape, as a basic eddy field. Next, we selected an eddy from the results of the following model and judged if it appeared in the base eddy field. We added it to the base eddy field if no eddy in the base eddy field had a profile that intersected with it. We iterated in this way until the results of all models were traversed. To guarantee that more well-shaped eddies were imported, we eliminated new eddies with fewer than four grid points.

Figure 4 shows the eddies detected by combining the results of multiple models. The upper panel shows a snapshot of the multi-model combination (CMM) method and the VG algorithm for eddy detection on 9 September 2016. On that day, the combination of multiple models detected 28 eddies, which was more than the 24 eddies detected by the VG algorithm. The most significant advantage of the combined multi-model results over the VG algorithm results was the detection of small eddies. CMM could more precisely find and characterize the boundaries of small eddies surrounding major eddies. The bottom panel compares the number of daily eddies detected by the CMM versus those identified by the VG and the deep learning model individually. The individual models that are given as comparisons only indicated the numbers of eddies discovered using the SLA and geostrophic velocity fields. The single deep learning model did not consistently outperform the VG algorithm in terms of daily detection efficiency, but the CMM found significantly

more eddies in each day's detection (an average of about 11 more eddies per day). In total, the CMM detected 103,437 eddies over ten years, which was well above the single-model maximum of 71,548.

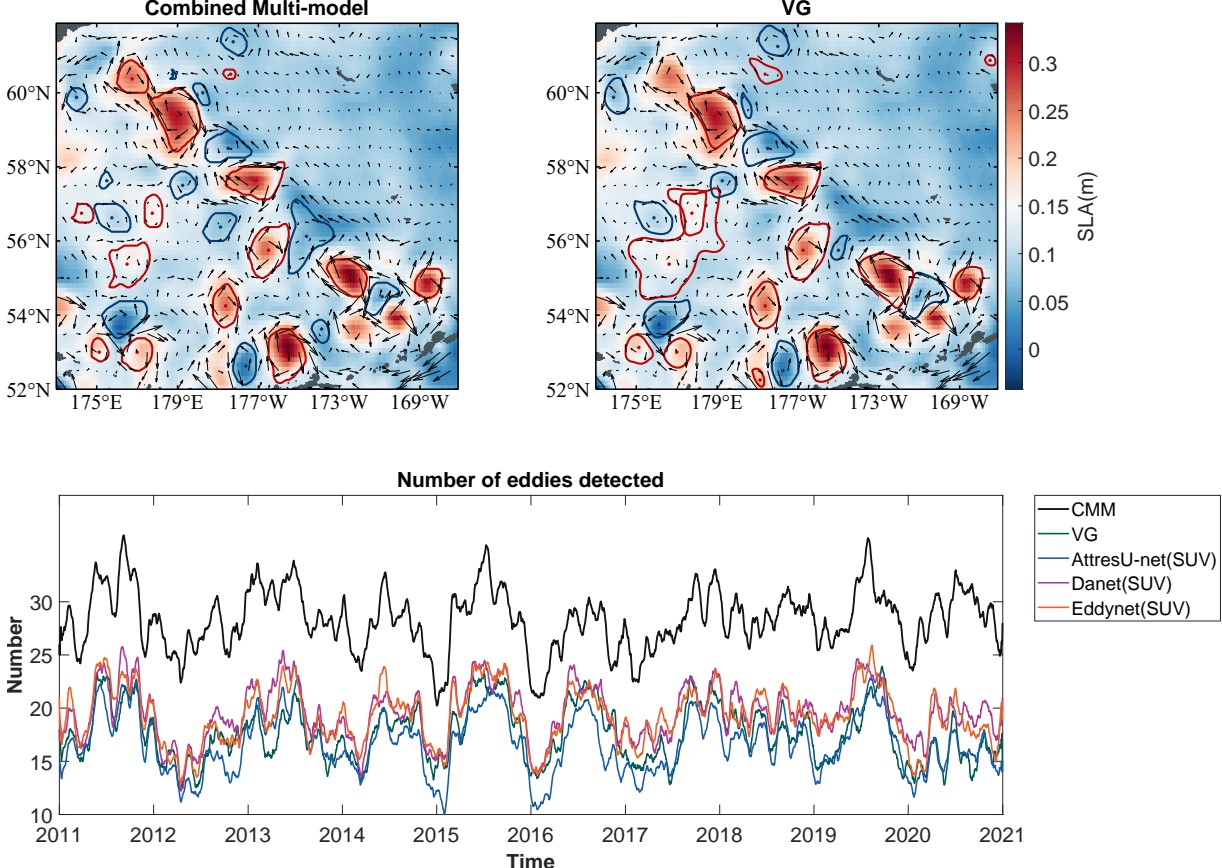

**Figure 4.** Combination of multi-model eddy detection results. The upper panel shows a snapshot (9 September 2016) of the comparison of multi-model recognition results with those of the VG algorithm. Red (blue) dots represent the center of an anticyclone (cyclone). The bottom panel shows the number of eddies detected by the CMM compared with the other algorithms in this paper. The curve represents the results after 31-day smoothing.

### 4.2. Validation

It is hard to find a method that can objectively assess the performance of eddy detection algorithms [20,33]. Effective evaluation of an eddy detection method is only possible once the true distribution of eddies has been determined. Therefore, referring to the validation process in Nencioli et al. [20], the same geometric constraints were used as guidelines for the construction of true eddy maps in this study. We randomly selected 10 days from 3653 test samples for both the CMM and VG algorithms to compare with the true eddy maps.

In order to display the results of the comparison and validate the efficiency of the two methods, the successful detection rate (*SDR*) and excess detection rate (*EDR*) were defined as follows [33]:

$$SDR = \frac{N_a}{N_t}, \tag{3}$$

$$EDR = \frac{N_e}{N_t}, \tag{4}$$

where $N_a$ is the number of eddies that are present in both the algorithm and the true maps; $N_e$ is the number of eddies considered by the algorithm to be eddies but not included in the true eddy maps; $N_t$ is the number of true eddies for a given day.

Tables 2 and 3 show our validation results. The average SDR of the CMM (69.37%) was much higher than that of the VG (38.69%). VG algorithms that worked well in other seas (>90%) [20] performed badly in high-latitude seas, revealing the difficulty of eddy detection in high-latitude seas. Consequently, the CMM algorithm is worthy of commendation as a result of its significant improvement in the SDR. Due to the lack of physical or geometric limitations, CMM eddies will have a few identifications (2.78%) in excess. In other research, this was not considered bad performance [20,33].

For example, Figure 5 shows a comparison between the true eddies and algorithm-detected eddies for day 2182 (21 December 2016). The green dots represent the true eddy distribution, while the green circles represent the eddies detected by the algorithm. As a result, dots enclosed by circles represent true eddies detected by the algorithm ($N_a$); hollow circles without kernels represent eddies detected by the algorithm as excess ($N_e$); the dots without circles are true eddies that the algorithm could not detect (missed eddies). The eddies missed by the CMM were mostly tiny, weak eddies, while VG missed several large eddies. The CMM had two over-detections. One occurred at an unclosed meander, while the other occurred at a border with low SLA. The missed eddies of the CMM were mostly due to the fact that the VG algorithm, using labeled data, was barely at the level of manual detection, resulting in some misdirection while also causing foreground object sparsity.

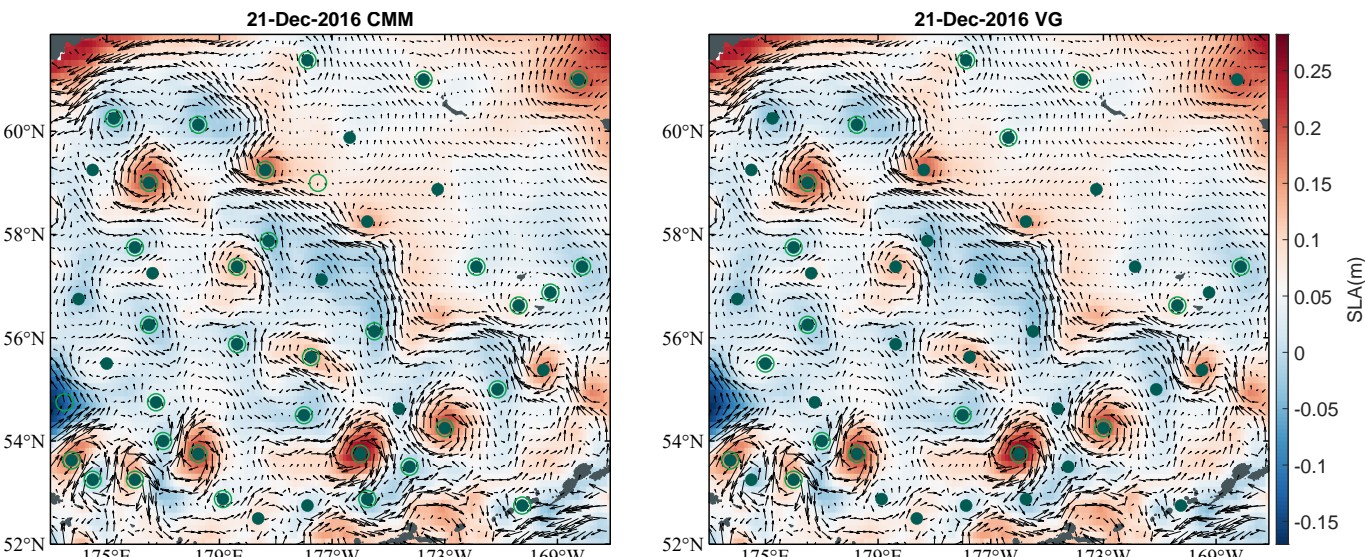

**Figure 5.** Comparison between true eddies (green dots) and algorithm-detected eddies (green circles) for day 2182 (21 December 2016). The left panel shows the multi-model detection results. The right panel shows the VG algorithm's detection results.

**Table 2.** CMM results for the 10 days used for the validation: True eddies are the eddies identified via manual detection; Na is the number of true eddies detected by the automated method; Ne is the number of eddy centers detected by the algorithm that do not correspond to a true eddy. SDR and EDR are defined in Equations (1) and (2).

| Day | 3379 | 2194 | 1981 | 3046 | 890 | 3631 | 129 | 1561 | 2182 | 752 | Total |
|---|---|---|---|---|---|---|---|---|---|---|---|
| True eddies | 42 | 39 | 39 | 39 | 47 | 45 | 51 | 46 | 44 | 39 | 431 |
| $N_a$ | 29 | 27 | 27 | 24 | 31 | 29 | 37 | 34 | 32 | 29 | 299 |
| $N_e$ | 1 | 2 | 1 | 0 | 1 | 2 | 1 | 2 | 2 | 0 | 12 |
| Missed eddies | 12 | 12 | 12 | 15 | 16 | 16 | 14 | 12 | 12 | 10 | 132 |
| SDR(%) | 69.05 | 69.23 | 69.23 | 61.54 | 65.96 | 64.44 | 72.55 | 73.91 | 72.73 | 74.36 | 69.37 |
| EDR(%) | 2.38 | 5.13 | 2.56 | 0.00 | 2.13 | 4.44 | 1.96 | 4.35 | 4.55 | 0.00 | 2.78 |

**Table 3.** VG results for the 10 days used for the validation.

| Day | 3379 | 2194 | 1981 | 3046 | 890 | 3631 | 129 | 1561 | 2182 | 752 | Total |
|-----|------|------|------|------|-----|------|-----|------|------|-----|-------|
| True eddies | 42 | 39 | 39 | 39 | 47 | 45 | 51 | 46 | 44 | 39 | 431 |
| $N_a$ | 17 | 14 | 19 | 14 | 19 | 17 | 13 | 21 | 17 | 15 | 166 |
| $N_e$ | 1 | 0 | 1 | 0 | 0 | 0 | 0 | 0 | 0 | 0 | 2 |
| Missed eddies | 23 | 25 | 20 | 24 | 28 | 28 | 38 | 25 | 25 | 24 | 260 |
| SDR(%) | 40.48 | 35.90 | 48.72 | 35.90 | 40.43 | 37.78 | 25.49 | 45.65 | 40.48 | 38.46 | 38.69 |
| EDR(%) | 2.38 | 0.00 | 2.56 | 0.00 | 0.00 | 0.00 | 0.00 | 0.00 | 0.00 | 0.00 | 0.47 |

*4.3. Eddy Size, Trajectory, and Lifetime*

Figure 6 shows the distribution of the number of eddy radii. The histograms of the eddy radii detected by the two methods—the CMM and VG algorithms—show a significant difference. This difference was most pronounced in the 0–25 km radius range, where CMM detected three times as many anticyclones and cyclones as VG. The detection efficiency of CMM was higher than that of the VG algorithm over most scale radii. Notable is the fact that the peak number of CMM detections (23,347 anticyclonic eddies and 20,951 cyclonic eddies) occurred in the 50–75 km radius range, while that of the VG algorithm occurs in the 25–50 km radius range (17,007 anticyclonic eddies and 17,183 cyclonic eddies). This made the frequency-skewed distribution of the CMM method's eddy radius more centered than the VG algorithm's. In addition, the histograms can also prove that the eddies in the land-slope area of the Bering Sea are generally small, with the largest anticyclonic eddies identified by the CMM during the decade having a radius of about 141 km and cyclonic eddies having a radius of about 121 km, while eddies with a radius greater than 200 km exist in the Kuroshio Extension [25].

The methods of observing eddies can be roughly grouped into two categories. The Eulerian method is concerned with spatial points and focuses on the eddies at each spatial location in the eddy field. The Lagrangian, on the other hand, is analyzed for individual eddies, with an emphasis on how each eddy in the eddy field changes over time. The deep learning models proposed in this study are all Eulerian in nature, and the above work detects how many eddies exist at any one time. If statistical information describing the motion of eddies is to be tallied, however, the outputs of the deep learning model must be further processed using Lagrangian methods. Taking into account the errors in altimetric data and the limitations of the detection method, a Lagrangian method was used to remove eddies with lifetimes of less than four weeks, resulting in a total of 742 eddy trajectories during 2011–2020, 383 of which were anticyclonic and 359 of which are cyclonic.

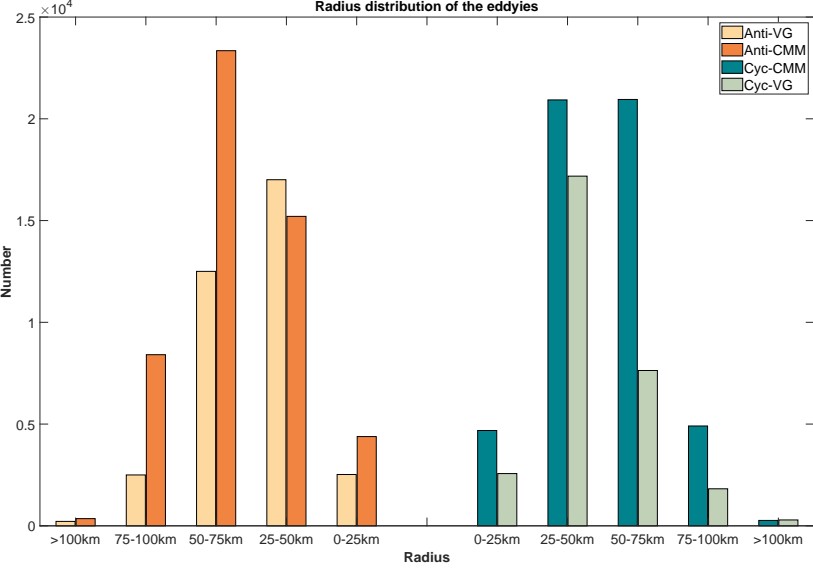

**Figure 6.** Radius distribution of the eddies detected with the two methods in the BSs during 2011–2020.

Figure 7 shows the eddy tracks with lifetimes longer than 4 weeks on the BSs. The majority of the eddies on the BSs migrate northwest following the slope current, with a few moving southwest towards the Bering Sea basin area. The eddies generated in the basin area near the slope current moved mainly westward and were confined within the basin. Most eddies in the BSs were generated in the BSs, and because these eddies could not go far, they tended to vanish near the slope current. The eddy motion detected by the VG method was similar to that of the CMM, but the sparse trajectories (361 in total) were insufficient for analyzing the eddy motion regularity of BSs.

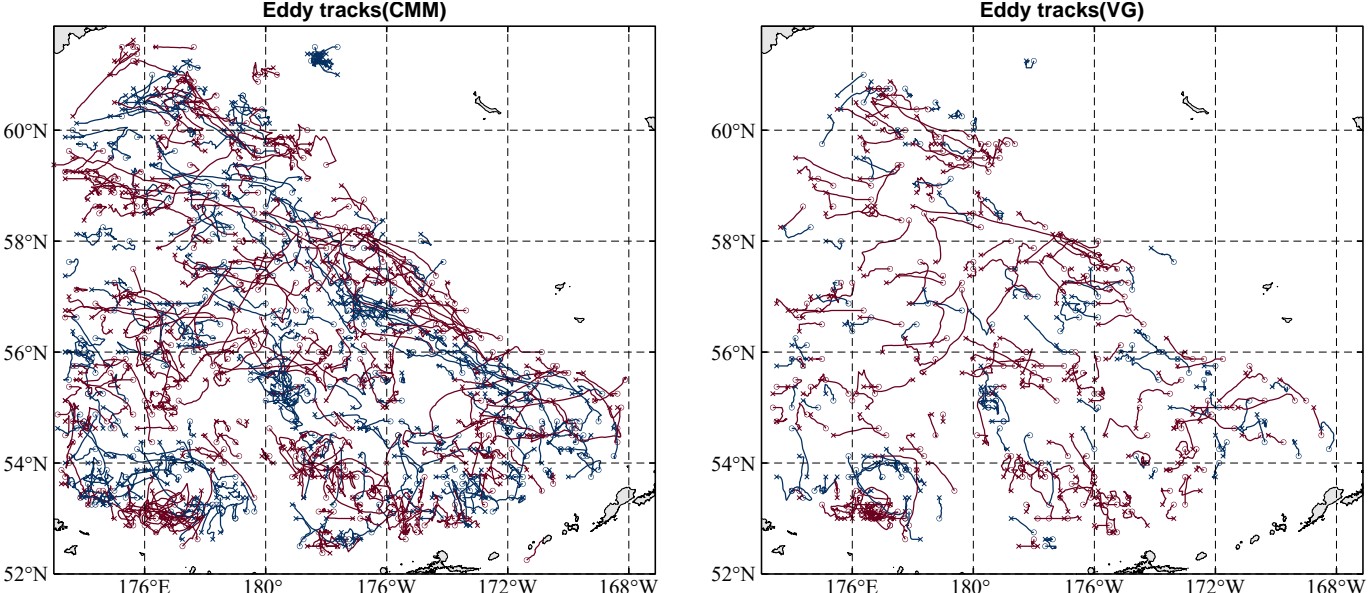

**Figure 7.** Eddy trajectories (for eddies with a lifetime of >4 weeks); red lines are trajectories of cyclonic eddies and blue lines are trajectories of anticyclonic eddies; the hollow circles indicate the starting positions of an eddy track, and the forks indicate the ending positions. The left panel is the eddy trajectory distribution of the CMM method. The right panel is the eddy trajectory distribution of the VG method.

Figure 8 shows the eddy lifetime distribution for lifetimes longer than 4 weeks. The lifetime distribution was based on the statistics of the eddy trajectory, which were taken from the results of the CMM and VG algorithms using the Lagrangian algorithm. The eddy lifetime distribution of the eddies discovered by the CMM is shown in the upper panel. Among the eddies detected by the CMM, the average lifetime of anticyclonic eddieswasis 10 weeks, the longest-lived anticyclonic eddy was 78 weeks, and the average lifetime of cyclonic eddies was 8 weeks, with the longest longevity being 29 weeks. It has been demonstrated in previous numerical and theoretical studies that anticyclonic eddies tend to have longer lifetimes than cyclonic eddies. The bottom panel shows the histogram of the eddy lifetime distribution identified by the VG algorithm. The VG algorithm was much less effective than the CMM method in locating smaller eddies. It was also unable to precisely track how eddies vanished; therefore, detected eddies were shorter-lived than those found using the CMM method (an average of 8 weeks and a maximum of 48 weeks for anticyclonic eddies and an average of 6 weeks and a maximum of 17 weeks for cyclonic eddies).

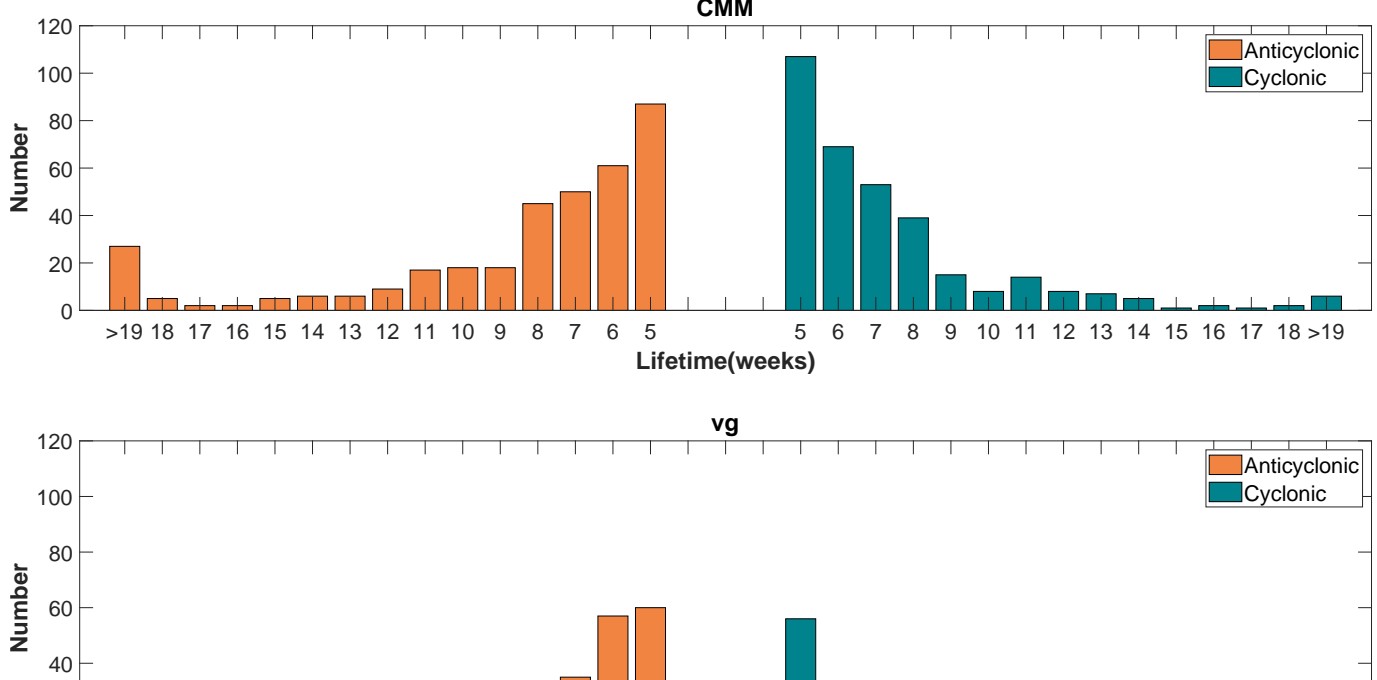

**Figure 8.** Lifetime distribution of the eddies detected by the two methods in the BSs during 2011–2020 (for eddies with a lifetime > 4 weeks). The upper panel is the lifetime distribution found with the CMM method. The bottom panel is the lifetime distribution found with the VG method.

## 5. Conclusions

An attention network comprised of a spatial attention module and a channel attention module may suppress feature responses in irrelevant background areas (such as non-eddy seas and land), thereby increasing the model's sensitivity to eddies with less computational effort. Geostrophic velocity data derived from the sea surface height data may reveal the geometric features of an eddy. Consequently, merging geostrophic velocity with SLA data to train deep learning models based on attention networks is suitable for high-latitude seas with more severe sample imbalance issues. This study used SLA and geostrophic velocity data from 1993 to 2010 together with eddy labels extracted by the VG method in the Bering Sea slope region as the training dataset and SLA and geostrophic velocity data from 2011 to 2020 as the test dataset. Not only were attention-network-based deep learning models compared to other deep learning models, but the results of models using both types of data were compared to those using SLA data alone as the training dataset. The deep learning algorithms detected more ocean eddies than the VG algorithm, with the attention-network-based Danet detecting the most eddies of the three deep learning models. Furthermore, integrating SLA data with geostrophic velocity data as model inputs improved the eddy detection results significantly over those obtained using solely SLA data. In this study, the outputs of multiple models were merged to enhance the daily eddy detection efficiency, and the number of discovered eddies was much greater than that of a single model or the VG method. Therefore, the multi-model combination method proposed in this paper for detecting high-latitude ocean eddies is important for the application of deep learning algorithms to satellite altimetry data.

**Author Contributions:** Conceptualization, Y.Z. and N.L.; methodology, Y.Z.; software, Z.Z., M.L. and L.F.; validation, Y.Z., Y.L. and H.C.; formal analysis, Y.Z., L.Y. and L.L.; investigation, All authors have read and agreed to the published version of the manuscript.

**Funding:** This research was funded by the National Science Foundation of China (No.42106232, No.41876222), Global Change and Air-Sea Interaction II (No.ZY0722031), and the Multidisciplinary Drifting Observatory for the Study of Arctic Climate (MOSAiC) (No.JD0619010).

**Data Availability Statement:** Data related to this paper can be downloaded from CMEMS (http://marine.copernicus.eu) (accessed on 6 August 2022). The codes used in the current analysis are available at https://github.com/mcyy6399/eddy_bss_detection.git (accessed on 6 August 2022).

**Acknowledgments:** Part of the code used for the networks of AttresU-net and Danet is publicly available and can be obtained from https://github.com/MoleImg/Attention_UNet.git (accessed on 6 August 2022) and https://github.com/niecongchong/DANet-keras.git (accessed on 6 August 2022).

**Conflicts of Interest:** The authors declare no conflict of interest.

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
