# Peer review of "Detection of Bering Sea Slope Mesoscale Eddies Derived from Satellite Altimetry Data by an Attention Network"

_remotesensing, doi:10.3390/rs14194974_

Round 1
Reviewer 1 Report
This article presents detection of Bering sea mesoscale eddies derived from satellite altimetry data by deep learning algorithm. This is well written paper. There is only one question for authours.
How do you know that eddies detected based on the deep learning algorithms are true eddies?
Author Response
Dear Reviewer 1, Please see the attachment.

Reviewer 2 Report
In this paper, the authors focus on application of machine learning algorithms to identify mesoscale eddies in the Bering Sea. The authors apply attention networks to satellite-derived sea level anomaly and geostrophic velocity datasets collected during 1993-2020. Then they detect eddies and provide some simple statistics about the obtained results (eddy radius, trajectories, lifetime). The paper addresses important physical processes in the Bering Sea and uses novel technique, which is a significant contribution to regional ocean science. However, there are certain important drawbacks of the study, which should be considered and reworked.
1. The authors state that the satellite-derived altimetry data used in the study has spatial resolution of 0.25° and daily temporal resolution. Moreover, the authors reduced the spatial resolution to 0.125° using nearest-neighbor interpolation. First, the applied straightforward reducing of spatial resolution to 0.125° seems to be incorrect. Why don’t you reduce spatial resolution even more, e.g., to 0.0125° using any kind of interpolation? How can you prove that the results remain correct for the reduced resolution? Second, the effective (real) spatial and temporal resolution of satellite altimetry products is much greater, please refer to Ballarotta et al. 2019 (Ocean Sci., 15, 1091–1109, 2019, https://doi.org/10.5194/os-15-1091-2019). As a result, even usage of spatial resolution of 0.25° and daily temporal resolution for machine learning algorithm should be validated against effective spatial and temporal resolution of satellite altimetry.
2. Please describe in detail the training/validation process of the eddy detection by machine learning. Was this processes performed by manual identification of eddies or any previously generated dataset of eddies was used?
Author Response
Dear Reviewer 2, Please see the attachment.

Reviewer 3 Report
This manuscript presented a method for detecting the Bering Sea slope oceanic eddy from satellite altimetry observations based the attention network. The authors constructed three models using SLA and geostrophic velocity data from 1993 to 2020 together with eddy labels extracted by the VG method in the Bering Sea slope region as the training dataset. According to experiments, several conclusions were drawn. Main conclusions include that deep learning algorithms discovered more ocean eddies than the VG method, with Danet, an attention network-based deep learning model, detecting the most eddies among the three deep learning models. The outputs of multiple models are merged to enhance the daily eddy detection efficiency, and the number of discovered eddies is much more than that of a single model or the VG method. Furthermore, the outputs of multiple models are integrated to improve the effectiveness of daily eddy detection, and the number of identified eddies is substantially higher than that of a single model or the VG approach.
The study produced good results and is reasonably well presented overall. It is commendable that the authors presented the problem of high-latitude eddy detection and proposed solutions with quite a lot of theoretical support and comparisons. However, because there are so many solutions, some of the fundamental theoretical underpinnings are not clearly provided, and there is insufficient discussion about model recognition's performance. So I suggest this manuscript could be accepted after moderate revision.
Detail comments:
LN21-22: “-through the east…” - “through the east pass…” would be more rigorous.
LN30-31: The last sentence is not suitable to put here in logical sense. It can be moved to somewhere in the front.
LN42-43: Rephase this sentence.
LN75-78: “To increase the number of eddy boundary pixels and improve detection accuracy…”- Is there any literature to support this?
LN136-139: It would be useful to provide specific formulas for these metrics.
LN145: No introduction about "The Eddynet model" can be found before.
LN156-157: Not rigorous enough. Can you provide some mathematical support?
Figure 3: No specification for abbreviations: 'Att','Dan','En'. The colors of contours are not obvious relative to the background.
Figure 4 caption: Is smoothing used for the time series? If so, I suggest stating the smoothing method.
Figure 4 caption: What do the red and blue dots represent?
LN190: Add detail desciption about the CMM method. How to combine different models.
Figure 6: I suggest to provide the trajectory figure of VG algorithm for comparison.
LN217-221: Supplement information of Eulerain method and Lagrangian method.
LN254: “-geostrophic velocity data from 2020…”- Here it is not consistent with the description in the data introduction section.
Author Response
Dear Reviewer 3, Please see the attachment.

Round 2
Reviewer 2 Report
The manuscript can be accepted in the current form
Author Response
Dear Reviewer 2, thank you for your approval.